# Predictive Risk Factors and Scoring Systems Associated with the Development of Hepatocellular Carcinoma in Chronic Hepatitis B

**DOI:** 10.3390/cancers16142521

**Published:** 2024-07-12

**Authors:** Ploutarchos Pastras, Evaggelos Zazas, Maria Kalafateli, Ioanna Aggeletopoulou, Efthymios P. Tsounis, Stavros Kanaloupitis, Konstantinos Zisimopoulos, Eirini-Eleni-Konstantina Kottaridou, Aspasia Antonopoulou, Dimosthenis Drakopoulos, Georgia Diamantopoulou, Aggeliki Tsintoni, Konstantinos Thomopoulos, Christos Triantos

**Affiliations:** 1Division of Gastroenterology, Department of Internal Medicine, University Hospital of Patras, 26504 Patras, Greece; ploutarchosp96@gmail.com (P.P.); zazasevangelos@gmail.com (E.Z.); mariakalaf@hotmail.com (M.K.); iaggel@upatras.gr (I.A.); makotsouno@gmail.com (E.P.T.); stavros5675@gmail.com (S.K.); condoczisimo@hotmail.com (K.Z.); elliekottaridi@gmail.com (E.-E.-K.K.); aspasia.antonopoulou@gmail.com (A.A.); dim.drak96@gmail.com (D.D.); geodiamant@hotmail.com (G.D.); kxthomo@hotmail.com (K.T.); 2Department of Internal Medicine, University Hospital of Patras, 26504 Patras, Greece; agtsintoni@gmail.com

**Keywords:** HBV, hepatocellular carcinoma, risk factors, GAG-HCC score, PAGE-B score

## Abstract

**Simple Summary:**

Chronic hepatis B still remains a global epidemic and one of the leading causes of hepatocellular carcinoma worldwide. We aimed to find out possible factors that could predict hepatocellular carcinoma development, as well as to compare the predictive performance of various, well-known, risk models in 632 patients with chronic hepatitis B. In our cohort, 34 patients developed hepatocellular carcinoma during follow-up. We found that patients that had cirrhosis, were males, had increased alcohol consumption and were older, had increased risk for cancer development in the multivariate analysis. Of the various risk models that were tested, Core promoter mutations and cirrhosis-HCC (GAG-HCC) and Platelet Age Gender–HBV (PAGE-B) risk models had the greatest performance; however, this performance significantly declined in the subgroup of patients with cirrhosis. In conclusion, certain baseline factors and risk models seem to predict the development of hepatocellular carcinoma in patients with chronic hepatitis B; however, this prediction declines in the subgroup of patients with cirrhosis and thus, we need novel risk models to be developed specifically for these patients.

**Abstract:**

Chronic hepatitis B (CHB) infection constitutes a leading cause of hepatocellular carcinoma (HCC) development. The identification of HCC risk factors and the development of prognostic risk scores are essential for early diagnosis and prognosis. The aim of this observational, retrospective study was to evaluate baseline risk factors associated with HCC in CHB. Six hundred thirty-two consecutive adults with CHB (*n* = 632) [median age: 46 (IQR: 24)], attending the outpatients’ Hepatology clinics between 01/1993–09/2020 were evaluated. Core promoter mutations and cirrhosis-HCC (GAG-HCC), Chinese University-HCC (CU-HCC), risk estimation for hepatocellular carcinoma in chronic hepatitis B (REACH-B), Fibrosis-4 (FIB-4), and Platelet Age Gender–HBV (PAGE-B) prognostic scores were calculated, and receiver operating curves were used to assess their prognostic performance. HCC was developed in 34 (5.38%) patients. In the multivariable Cox regression analysis, advanced age (HR: 1.086, 95% CI: 1.037–1.137), male sex (HR: 7.696, 95% CI: 1.971–30.046), alcohol abuse (HR: 2.903, 95% CI: 1.222–6.987) and cirrhosis (HR: 21.239, 95% CI: 6.001–75.167) at baseline were independently associated with the development of HCC. GAG-HCC and PAGE-B showed the highest performance with c-statistics of 0.895 (95% CI: 0.829–0.961) and 0.857 (95% CI: 0.791–0.924), respectively. In the subgroup of patients with cirrhosis, the performance of all scores declined. When treated and untreated patients were studied separately, the discriminatory ability of the scores differed. In conclusion, HCC development was independently associated with advanced age, male sex, alcohol abuse, and baseline cirrhosis among a diverse population with CHB. GAG-HCC and PAGE-B showed high discriminatory performance to assess the risk of HCC development in these patients, but these performances declined in the subgroup of patients with cirrhosis. Further research to develop scores more specific to certain CHB subgroups is needed.

## 1. Introduction

Hepatocellular carcinoma (HCC) is the second most common cause of cancer-related deaths on a global scale, leading to more than 800,000 deaths per year [1]. During the last few decades, an increase in both incidence and mortality from HCC has been observed worldwide [1]. Chronic hepatitis B (CHB), attributed to hepatitis B virus (HBV) infection, is one of the leading causes of HCC worldwide, together with chronic hepatitis C virus (HCV) infection, alcohol consumption and aflatoxin exposure. In African and East Asian countries, CHB is the major etiological factor accounting for 60% of HCCs, whereas in Europe it attributes for 20% of cases while chronic HCV infection predominates [2]. In Greece, chronic viral hepatitis and mainly HBV is still the leading cause of HCC, although a trend towards an increase in the prevalence of HCV and non-alcoholic fatty liver disease together with a decrease in the prevalence of CHB has been observed in the period after 2000 [3]. Chronic HBV infection is a dynamic entity, and its natural history depends on the interplay between viral replication and the host immune response. Through this interaction, it can progress to cirrhosis and HCC development. It is widely known that cirrhosis is a major risk factor for HCC occurrence and the incidence of HCC per year in patients with CHB and cirrhosis is estimated at approximately 2–5% [2]. However, in CHB, HCC can be developed independently of the presence of cirrhosis, and even in treated patients with long-term suppression of HBV DNA levels.

Several factors have been identified as possessing prognostic significance in the development of HCC in CHB, either related to the host including cirrhosis, increased necro-inflammatory activity, male sex, advanced age, active smoking, co-existence of metabolic syndrome, alcohol abuse or diabetes mellitus, and chronic co-infection with other hepatitis viruses (HDV, HCV, human immunodeficiency virus (HIV)), or to viral characteristics such as HBV genotype C, high HBV DNA and/or HbsAg levels and core promoter mutations [4]. For this purpose, various predictive scores have been developed [4,5,6]. Some of them, i.e., guide with age, gender, HBV DNA, core promoter mutations and cirrhosis-HCC (GAG-HCC), Chinese University-HCC (CU-HCC), and risk estimation for hepatocellular carcinoma in chronic hepatitis B (REACH-B), have been validated in Asian untreated cohorts and have varying performance in Caucasian patients. Platelet Age Gender–HBV (PAGE-B) is a prognostic scoring model that incorporates platelet count, age, and sex and has been validated in Caucasian patients with CHB under antiviral treatment with tenofovir or entecavir [6].

The aim of this retrospective, observational study was to evaluate the prognostic significance of various baseline factors for HCC development in consecutive Caucasian patients with CHB residing in South-Western Greece, as well as to evaluate the performance of GAG-HCC, CU-HCC, REACH-B, Fibrosis-4 (FIB-4), and PAGE-B in this real-world setting.

## 2. Materials and Methods

### 2.1. Data Collection and Selection of Participants

In total, seven hundred seventy consecutive Caucasian adults (*n* = 770) mono-infected with chronic HBV infection (HbsAg positive for more than 6 months) attending the Hepatology outpatient clinics of the University Hospital of Patras in Greece between January 1993 and September 2020 were retrospectively evaluated. Patients who had HCC on baseline or other comorbidities including HCV or hepatitis D virus infection, Wilson’s disease, primary biliary cholangitis, autoimmune hepatitis, or non-alcoholic steatohepatitis were excluded from the analysis as well as those with missing follow-up data or not of Caucasian origin (excluded: occult HBV infection: 111, chronic HCV = 2 patients, HCC at baseline = 8, no follow-up data = 17). Finally, 632 patients were included in the analysis.

Eligibility criteria for CHB were determined by one of the potential combinations of serological markers (HbsAg, HbeAg/anti-Hbe, anti-HBc IgM, HBV DNA) according to the Center for Disease Control and Prevention (CDC) [7,8] and CHB was diagnosed according to the European Association for the Study of the Liver (EASL) guidelines [2].

Patients consisted of a heterogeneous (untreated and treated/cirrhotic and non-cirrhotic) group of adults chronically infected with HBV; thus, the only outcome that was assessed was HCC development, and any comparisons performed were between participants with or without HCC-HBV.

### 2.2. Ascertainment of Outcome, Risk Factors and Prognostic Scores

HCC diagnosis was based on non-invasive imaging criteria obtained by contrast-enhanced computed tomography (CT) or dynamic magnetic resonance imaging (MRI) and/or pathology according to existing guidelines [1]. The imaging criteria consisted of the identification of typical imaging hallmarks of HCC such as the hypervascularity in the arterial phase with washout of intravenous contrast in the portal venous and/or delayed phases [1,9]. Liver cirrhosis was diagnosed based on physical examination, clinical biomarkers, transient elastography, liver biopsy or a combination of these [10].

The patients were followed up for a mean of 49 months (95% CI: 45–55 months) until death or liver transplantation.

Several potential risk factors for HCC development were considered, such as age, sex, smoking, family history of HCC, ascites, body mass index (BMI), cirrhosis, and alcohol abuse, and laboratory tests were consecutively recovered from the review of medical files and included in the analysis. The most-used risk scoring systems of HCC in patients with CHB were calculated. More specifically, the PAGE-B, the GAG-HCC, the CU-HCC, the FIB-4, and the REACH-B scores were evaluated [11]. The CU-HCC components are age, albumin and bilirubin values, HBV DNA copies, and cirrhosis, and range between 0 and 44.5 [12]. The GAG-HCC score also includes gender and basal core promoter mutations, but it excludes albumin, bilirubin, and HBV DNA of the CU-HCC components [13]. Because of the possible non-availability of the core promoter mutations, the GAG-HCC score has been simplified to exclude the parameter of mutations [14]. On the other hand, REACH-B contains gender, age, alanine aminotransferase (ALT) value, HbeAg presence, and HBV DNA copies, while PAGE-B estimates age, gender, and platelets [15,16]. The FIB-4 score has been used in limited ethnic groups as a model of risk assessment for HCC development, and comprises age, aspartate aminotransferase (AST), ALT, and platelets [17].

### 2.3. Ethics

The study protocol was reviewed by the local institutional review board of the University Hospital of Patras in Greece which approved the study as meeting national and international guidelines for medical research on humans (ethical approval number: 13/10.04.2019).

### 2.4. Statistical Analysis

Continuous variables were expressed as medians and interquartile ranges (IQR), while categorical variables were presented as counts and corresponding percentages. Pearson’s chi-square test or Fisher’s exact test, if applicable, were used to identify potential differences between categorical variables. The Shapiro–Wilk test was utilized to assess the normality of the continuous variables. As the data did not follow a normal distribution, the comparison of median values between the two groups (HCC versus non-HCC) was performed using the non-parametric Mann–Whitney U test. Baseline characteristics and laboratory values were recorded for all patients at their first clinical visit to our center, and the corresponding predicting scores were calculated. We used the Cox proportional hazards model to perform a time-to-event analysis in order to assess probable factors associated with the development of HCC among patients with HBV. Data for patients who did not develop HCC were censored at death, liver transplantation, or the last clinical visit. Non-proportionality was tested by creating and analyzing partial residual plots for each variable (Schoenfeld residual proportional hazards test). We utilized a linear regression model, and the Variance Inflation Factor (VIF) was evaluated to test for multicollinearity. To avoid simultaneous inclusion of highly correlated variables, we set a threshold of VIF > 2.5 as not acceptable [18]. All variables of interest with a *p*-value of <0.05 in the univariate analysis were first included in the multivariate Cox model and then eliminated using backwards selection. Receiver operating curves (ROCs) were calculated for all predicting scores, and the corresponding areas under the ROC (AUROC) were determined to assess their prognostic performance. Subgroup analyses were conducted based on cirrhosis status. For the better performing predictive scores, Kaplan–Meier analysis and log-rank test were performed to estimate and compare, respectively, the HCC-free curves between low- and high-risk patients. The statistical significance level was set to 5% (*p* < 0.05). Statistical analysis was performed using the IBM SPSS version 26.0.

## 3. Results

### 3.1. Baseline Characteristics of Participants

The baseline characteristics of the 632 included patients are presented in Table 1. The median age of participants was 46 (IQR: 24) and 384 (60.8%) were male. Cirrhosis was present in 75 (11.9%) patients at baseline.

### 3.2. Outcomes during Follow-Up

All patients were followed up for a mean of 49 months (95% confidence interval (CI): 45–55 months). Two hundred and five (32.4%) patients received anti-HBV treatment [interferon (IFN)-based: N = 72 (35.1%) or nucleos(t)ide analogs (NUCs): 133 (64.9%)] in a median of 3 months (range 0–213) following inclusion in the study.

Thirty-four patients (5.38%) developed HCC during follow-up. Follow-up for patients who remained free of HCC was 48 months (95% CI: 44–53 months), whereas the time to HCC development in patients who developed HCC was 59 months (95% CI: 39–79 months) (*p* = 0.864). During this follow-up period, seven patients (1.1%) died, six due to liver-related causes. Among those who developed HCC, four patients (11.8%) died due to liver-related causes, compared to two patients (0.3%) among those who remained HCC-free (*p* < 0.001).

### 3.3. Comparison of Clinical Characteristics and Laboratory Values between HCC and Non-HCC Cohorts

The main characteristics of the patients with and without HCC and the comparison between them are summarized in Table 2. Patients that developed HCC more frequently had these characteristics: males (*p* < 0.001), old (*p* < 0.001), with cirrhosis (*p* < 0.001), ascites (*p* = 0.035) and esophageal varices (*p* < 0.001), and with a history of alcohol abuse (*p* < 0.001) at baseline. Furthermore, they more frequently received antiviral treatment (*p* < 0.001) and they presented lower platelet (PLT) (*p* < 0.001) and albumin values (*p* = 0.002) and higher total bilirubin (*p* = 0.002), INR, SGOT, SGPT, GGT, ALP (all *p* < 0.001), and creatinine levels (*p* = 0.036). CP and MELD scores at baseline were higher in patients that developed HCC as well as GAG-HCC, CU-HCC, REACH-B, FIB-4, and PAGE-B scores (all *p* < 0.001).

### 3.4. Univariate and Multivariate Analysis for HCC Development

A univariate analysis was performed exploring the variables associated with HCC development (Table 3). Age, gender, cirrhosis at baseline, alcohol abuse, platelets, INR, and total bilirubin and albumin were the factors related to HCC development in unadjusted univariate analysis. In the multivariate analysis, advanced age (HR: 1.086, 95% CI: 1.037–1.137, *p* < 0.001), male sex (HR: 7.696, 95%CI: 1.971–30.046, *p* = 0.003), alcohol abuse (HR: 2.903, 95%CI: 1.222–6.987, *p* = 0.016), and cirrhosis at baseline (HR: 21.239, 95%CI: 6.001–75.167, *p* < 0.001) were found to be independent predictors of HCC development (Table 3).

### 3.5. Performance of Risk Scores for Development of HCC

ROC curves for CU-HCC, GAG-HCC, PAGE-B, REACH-B, and FIB-4 scores were constructed to predict the HCC outcome in all patients at baseline (Figure 1).

The largest under-the-line area was delimited by GAG-HCC’s line (c-statistic = 0.895, 95%CI = 0.829–0.961, *p* < 0.001). The following larger area belonged to PAGE-B (c-statistic = 0.857, 95%CI = 0.791–0.924, *p* < 0.001). The remaining scores, namely CU-HCC, FIB-4, and REACH-B presented similar but moderate prognostic values (Table 4).

The cutoff points for each score that maximized sensitivity and specificity were calculated and the results are presented in Table 5.

In the subgroup of patients that did not receive antiviral treatment, all scores showed similar c-statistics with a high prognostic value. On the other hand, in the group of patients that received antiviral treatment, only GAG-HCC (c-statistic: 0.846, 95%CI: 0.75–0.943, *p* < 0.001) and PAGE-B (c-statistic: 0.818, 95% CI: 0.728–0.909, *p* < 0.001) had a good discriminatory performance (Table 4, Figure 2a,b). The performance metrics for each prognostic score in the subgroups of patients that received or did not receive antiviral treatment are presented in Appendix A.

We then conducted a subgroup analysis based on cirrhosis status (Table 4). In patients with cirrhosis at presentation, the performance of all scores declined. More specifically, GAG-HCC (c-statistic: 0.599, *p* = 0.330), PAGE-B (c-statistic: 0.602, *p* = 0.315), and REACH-B (c-statistic: 0.565, *p* = 0.523) showed a modest performance, whereas FIB-4 (c-statistic: 0.244, *p* = 0.012) and CU-HCC (c-statistic: 0.407, *p* = 0.361) scored rather low (Figure 3a).

In the group of patients without cirrhosis, GAG-HCC (c-statistic: 0.816, 95% CI: 0.685–0.947, *p* = 0.008) and PAGE-B (c-statistic: 0.808, 95% CI: 0.708–0.908, *p* = 0.010) showed greater predictive performance compared to the other scores (Figure 3b).

The performance metrics for each risk score in the subgroups of patients who were cirrhotic/non-cirrhotic are presented in Appendix A.

In Kaplan–Meier analysis for the highest scoring models (GAG-HCC and PAGE-B) the observed cumulative probability of HCC development using the established cut-offs for these scores increased for GAG-HCC ≥ 101 and PAGE-B > 12, respectively (Figure 4a,b).

## 4. Discussion

The development of HCC in CHB is a multi-step complex process affected by a variety of risk factors. The aim of the present study was to assess potential baseline clinical and laboratory factors associated with HCC development and to evaluate the predictive performance of commonly used risk stratification scores of HCC in a large diverse cohort of Caucasian patients followed up for almost 20 years. Our findings indicate that older age, male gender, alcohol abuse, and cirrhosis at baseline can predict the incidence of HCC in patients with HBV and that GAG-HCC and PAGE-B scores are the most reliable systems to predict this outcome in this cohort of patients with CHB.

On a cellular level, HBV itself has high carcinogenic potential and may lead to HCC development through several oncogenic pathways, mostly through HBV DNA integration to the host genome leading to genomic instability and insertional mutations, and through the production of various viral proteins [4]. In previous studies, a possible correlation between HBV-associated mutations (T1762/A1764 and Pres detection) and HCC has been observed [19,20]. In addition, the “viral oncoprotein” Hbx protein affects DNA repair and the control of cell growth [21]. The presence of necro-inflammation enhances this premalignant microenvironment through the continuous cycle of hepatocyte death and regeneration, and all these explain why HCC can develop even in the absence of advanced fibrosis [1,22,23]. However, cirrhosis is undoubtedly one of the major contributors to hepatocarcinogenesis, and this is reflected by the increased frequency of HCC development in patients with cirrhosis [24], a finding also confirmed in our study: patients with cirrhosis had a more than 20 times higher risk of developing HCC in the multivariate analysis. In discordance with other reports [25,26,27], we did not find any association between the risk of HCC development and HBV viral load; however, there are other relevant studies that also failed to show an association between HBV DNA or other HBV serum markers and this outcome [24]. It seems that there is a dose–response relationship between the HBV viral load and HCC, and this is more prominent for patients who are seronegative for HbeAg with normal ALT and absence of baseline cirrhosis [27]. More importantly, baseline HBV DNA levels consistently fail to correlate with HCC risk in the group of patients under treatment with nucleoside analogues [28]. Our cohort included a rather heterogenous group of patients with HBV and with or without cirrhosis both treated and untreated and this might explain the lack of association between HCC development and HBV DNA levels in our study. Nevertheless, the threshold of baseline HBV viral load that predicts HCC is not elucidated yet and needs to be further investigated.

Male sex, increased alcohol consumption, and advanced age are widely accepted host risk factors for HCC progression in CHB in the majority of studies [2,25,29,30], not only in the untreated but also in patients under treatment with nucleos(t)ide analogues. We confirmed these widely accepted findings as all three were found significant in the multivariate analysis. The male dominance in HBV-related HCC is estimated to be 2.9:1 [31], implying a possible role for sex hormones in hepatocarcinogenesis. In a meta-analysis [32] of 66 longitudinal studies and randomized controlled trials including 347859 untreated patients with HBV, increasing age [RR 1.7 (95% CI: 1.4–2.1) for 10-year increase], heavy alcohol intake [≥60 g/dL, (RR 2.1 (95% CI: 1.4–4.6)], and male gender (RR 2.7, 2.1–3.3) were positively associated with HCC incidence. In another Chinese meta-analysis of 27 studies (3156 Chinese HBV cases), alcohol consumption had a pooled odds ratio of 2.19 (95% CI: 1.53, 3.13) for HCC development [25], similar to our study.

We failed to find thrombocytopenia as a significant independent prognostic factor for HCC development, although it achieved statistical significance in the univariate analysis. It is well known that the platelet count is reduced as liver disease progresses and as portal hypertension establishes [24]. Papatheodoridis et al. were the first to observe that low platelets were a significant individual HCC risk factor in treated Caucasian patients with HBV with or without cirrhosis [24] and, thus, they incorporated the platelet count together with age and gender to develop the PAGE-B score [6]. We attribute this discordance to the diverse population of our study and the small proportion of patients that developed HCC (5%) during follow-up.

Among the five HCC risk scores that were evaluated in our cohort, only PAGE-B and GAG-HCC showed high discriminative ability to predict HCC development. The others, namely FIB-4, REACH-B, and CU-HCC, had moderate accuracy. In the subgroup analysis according to treatment status, all five HCC risk scores scored highly in the untreated sub-population, but when tested in the treated cohort their performance declined except for GAG-HCC and FIB-4 which remained high. This is in accordance with the results of previous studies. As mentioned before, GAG-HCC, REACH-B, and CU-HCC were first developed in cohorts of untreated Asian patients with CHB. They have varying performance in Asian populations with treated and untreated CHB [5,33], but they have repeatedly scored moderate to low when tested in Caucasian populations, especially in treated patients [24,34,35]. More specifically, in the study by Papatheodiridis et al. including 1666 patients treated for HBV [24], none of these scores were found to be an independent predictor for HCC in the multivariate analysis (although significant in the univariate), and their performance was poor to moderate with a c-index of 0.76, 0.62, and 0.61 for GAG-HCC, CU-HCC, and REACH-B, respectively. In another study by Arends et al. [34] of 744 patients with CHB treated with entecavir, the c-statistics for the Caucasian subgroup (42%) were 0.66, 0.74, and 0.54, for CU-HCC, GAG-HCC, and REACH-B, respectively, and their diagnostic performance declined during treatment with entecavir. Unexpectedly, in our cohort GAG-HCC had a high performance with a c-statistic of 0.895 that is not in accordance with the abovementioned studies. However, the heterogeneity of our population might again explain this discordance. In support of this hypothesis, in a study by Brouwer et al. [16] with a similar design to our study which included 557 patients with CHB (47% Caucasians) both treated and untreated, GAG-HCC scored highly with a c-statistic of 0.91 (95% CI 0.86–0.96). Furthermore, we should also note that, as in other studies, GAG-HCC scored higher than REACH-B and CU-HCC; similarly, in our study the superiority of the GAG-HCC score is depicted.

All these three models (GAG-HCC, REACH-B, and CU-HCC) include baseline HBV DNA levels in their formulas; however, as mentioned before, there is an uncertainty in the literature when incorporating pre-treatment viral load in the HCC risk scores, especially regarding patients treated for HBV [35]. Currently, there is an effort to develop predictive models that do not incorporate variables modifiable by antiviral therapy, such as PAGE-B [6]. PAGE-B has a high performance in the setting of HCC prediction both in treated [6] and untreated patients [16] and this was also depicted in our study with a c-statistic of 0.856 that remained high when treated and untreated patients were studied separately. Our results are in accordance with the study by Brouwer et al. [16] where PAGE-B was the overall best performing risk score for all outcomes (both survival and HCC development) compared to REACH-B, FIB-4, AST to Platelet Ratio Index (APRI), and GAG-HCC and CU-HCC risk models, with a c-statistic of 0.91 (95% CI 0.82–0.99).

Another interesting finding of our study was the significant decline in the diagnostic performance of all HCC risk scores in the subgroup of patients with cirrhosis. This has been also confirmed in the study by Brouwer et al. [16] where HCC risk scores were lower for the prediction of any clinical outcome in patients with advanced fibrosis. Furthermore, Arends et al. [34] observed that when the discriminatory performance of CU-HCC, GAG-HCC, and REACH-B was tested in the subgroups of patients with and without cirrhosis, only GAG-HCC remained predictive for HCC occurrence. These observations together with the results of our study imply the need for specific HCC risk scores depending on the different subgroups of CHB patients.

Some limitations of the present study should be acknowledged. Despite the large cohort of patients, the retrospective single-center design has a certain bias including handling of the missing data. Furthermore, as an observational study, it could not avoid residual confounding. However, we extensively reviewed the municipal record database, patients were followed up for a rather long period and we included patients who were mono-infected with available follow-up data. Lastly, we failed to obtain data on HbsAg levels and transient elastography values for all patients and, thus, these variables were not tested as potential HCC predictors.

## 5. Conclusions

In conclusion, male sex, advanced age, increased alcohol consumption, and baseline cirrhosis were significant individual independent predictors of HCC development in our cohort, confirming the results of previous studies. Among the HCC risk scores, PAGE-B and GAG-HCC had the highest discriminative ability to predict HCC in this diverse Caucasian population with HBV, but this declined in the group of patients with cirrhosis. There is ongoing research to identify the best-performing risk model from those already developed, as well as to create others specifically validated for certain subgroups of patients with CHB (treated vs. untreated, advanced fibrosis vs. non-advanced fibrosis, different ethnic origin etc.). The application of HCC prediction models for risk stratification will provide individualized HCC surveillance in patients with CHB. Moreover, the HCC risk models could expand the existing criteria for the initiation of antiviral treatment [2] based on validated cut-offs above which there will be a substantial increase in HCC risk. Thus, the conduct of further studies is deemed necessary in order to also elucidate the biological pathways underlying the correlation of HCC development with these risk factors.

## Figures and Tables

**Figure 1 cancers-16-02521-f001:**
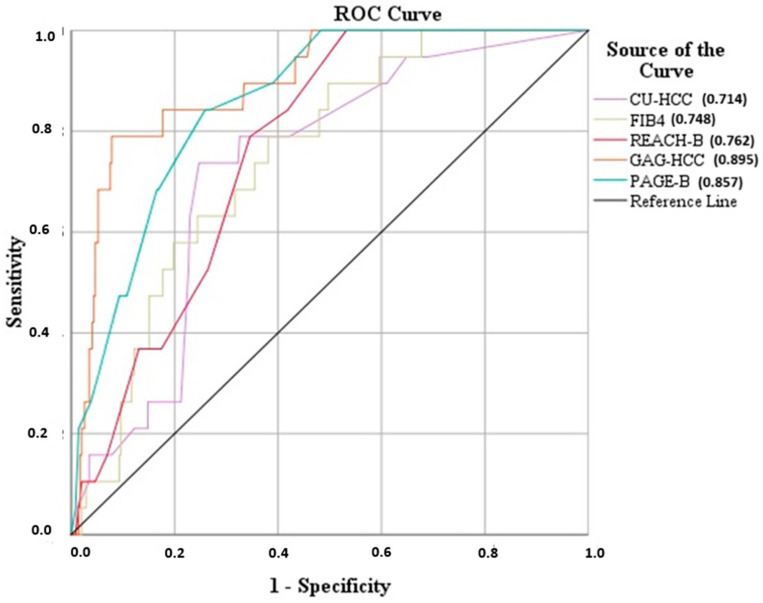
ROC curve analysis for the assessment of the prognostic value of PAGE-B, Reach-B, FIB-4, CU-HCC, and GAG-HCC in the total population.

**Figure 2 cancers-16-02521-f002:**
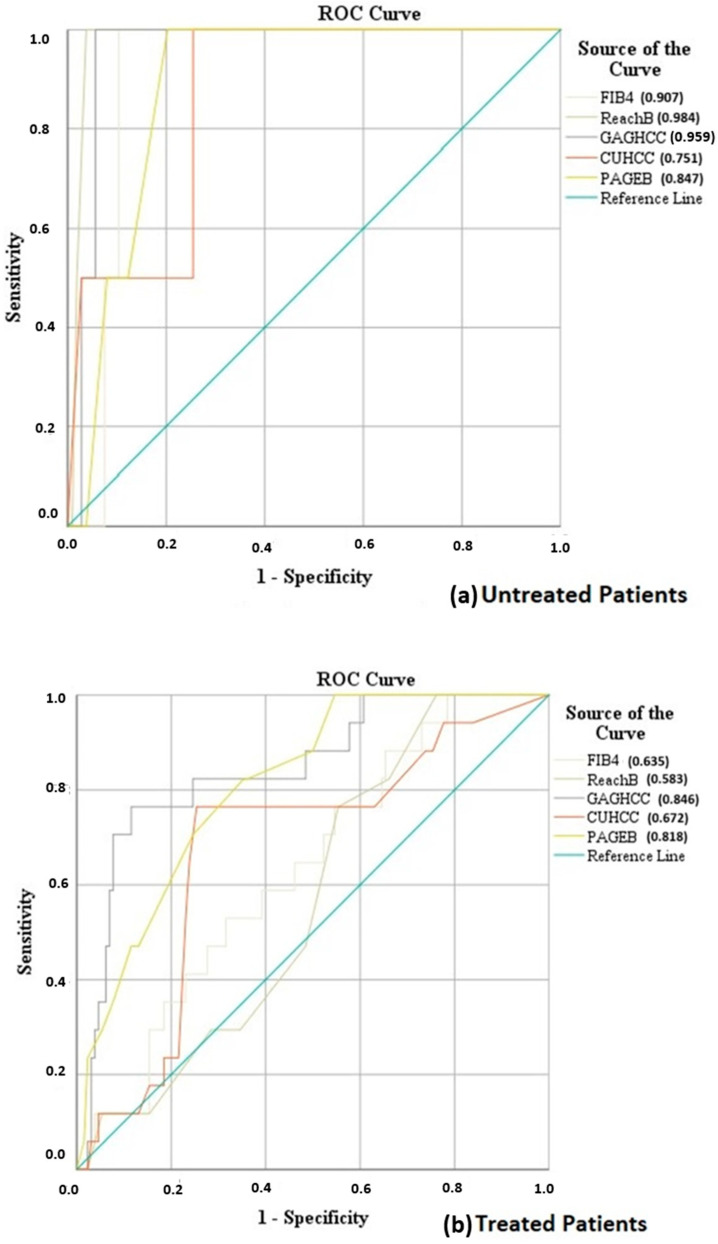
ROC curve analysis for the assessment of the prognostic value of PAGE-B, Reach-B, FIB-4, CU-HCC, GAG-HCC in (**a**) patients that did not receive antiviral treatment and (**b**) patients treated with antiviral therapy.

**Figure 3 cancers-16-02521-f003:**
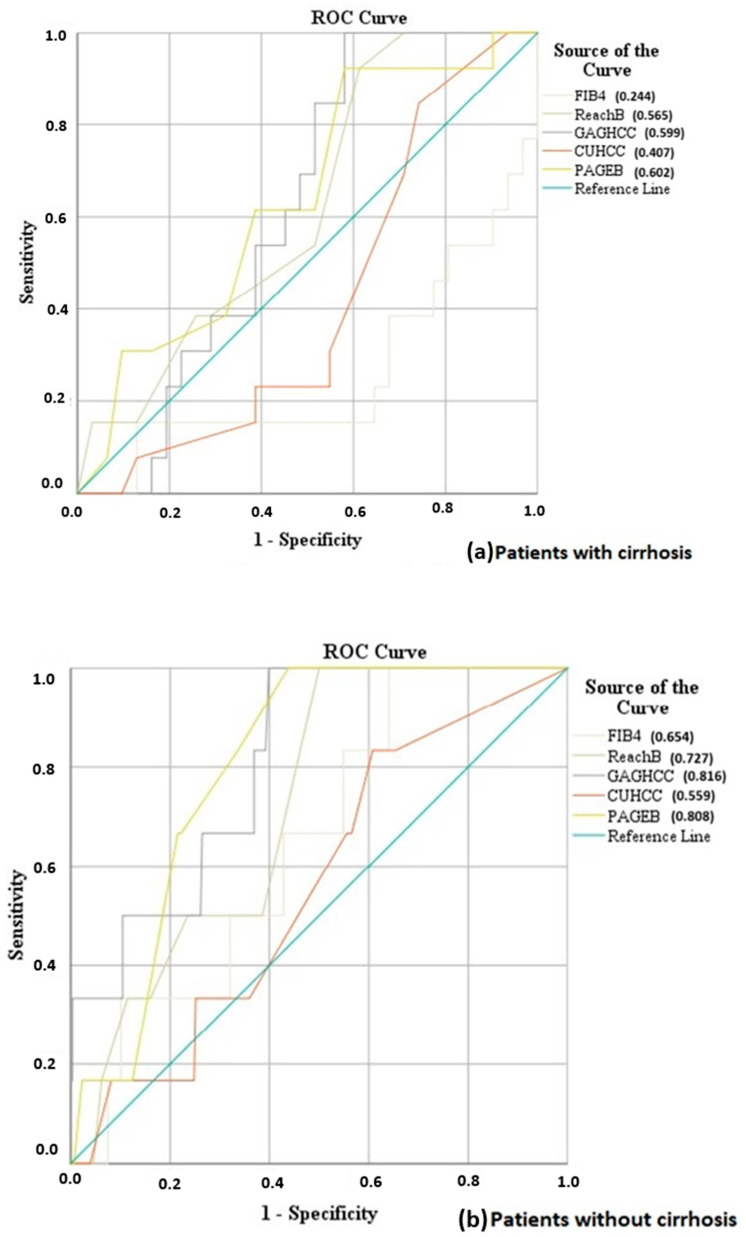
ROC curve analysis for the assessment of the prognostic value of PAGE-B, Reach-B, FIB-4, CU-HCC, GAG-HCC in the subgroups of (**a**) patients with cirrhosis and (**b**) patients without cirrhosis.

**Figure 4 cancers-16-02521-f004:**
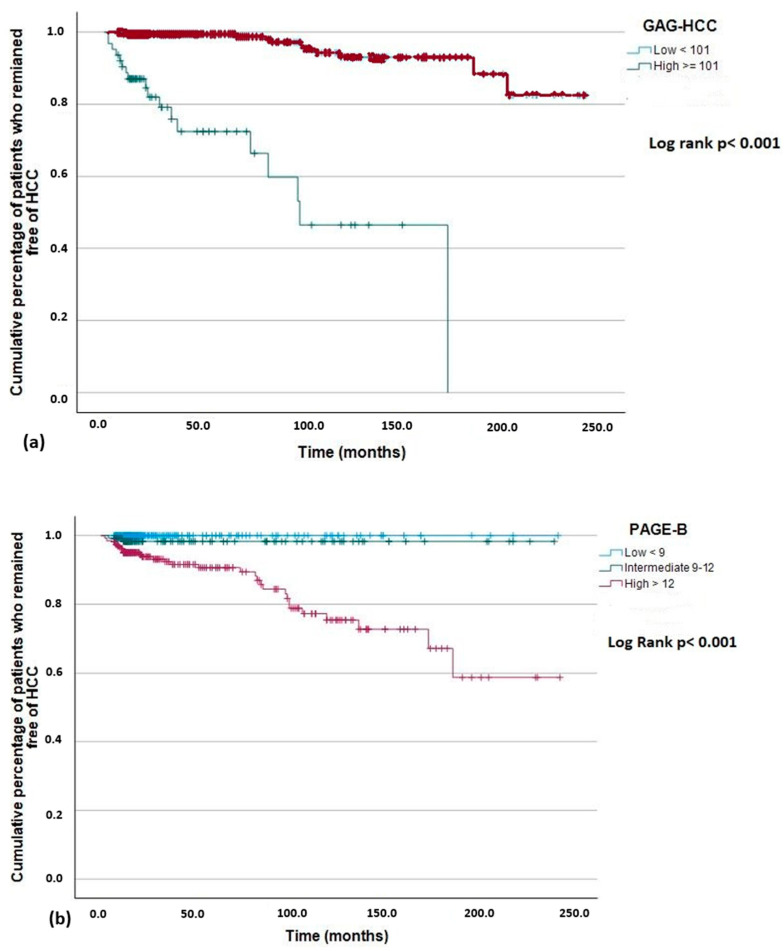
Kaplan–Meier curves for the cumulative probability of hepatocellular carcinoma (HCC) development according to (**a**) GAG-HCC and (**b**) PAGE-B established cut-offs.

**Table 1 cancers-16-02521-t001:** Baseline characteristics of the study population.

**Baseline Characteristics**	**N (%)**
Gender, male	384 (60.8%)
Active smoker	186 (29.4%)
Alcohol abuse	112 (17.7%)
Family History of HCC	15 (2.4%)
Cirrhosis	75 (11.9%)
CP Stage A	51 (68.0%)
CP Stage B	22 (29.3%)
CP Stage C	2 (2.7%)
Ascites	24 (3.8%)
Esophageal Varices	37 (5.9%)
Portal Gastropathy	26 (4.1%)
Portal Vein Thrombosis	1 (0.2%)
Hepatic Encephalopathy	0 (0.0%)
Phase of Infection	
HbeAg (+) Chronic Infection	4 (0.6%)
HbeAg (+) Chronic Hepatitis	38 (6.0%)
HbeAg (-) Chronic Infection	319 (50.5%)
HbeAg (-) Chronic Hepatitis	271 (42.9%)
**Baseline Characteristics**	**Median (IQR)**
Age (years)	46.0 (34–60)
BMI	26.1 (23.6–28.6)
MELD score	10.0 (8–12.5)
GAG-HCC	56.0 (43–71)
CU-HCC	3.0 (0–20)
REACH-B	7.0 (4–10)
FIB-4	1.11 (0.72–2.12)
PAGE-B	10.5 (6–16)

Abbreviations: N, number; HCC, hepatocellular carcinoma; CP, Child–Pugh; IFN, interferon; NUCs, nucleos(t)ide analogs; IQR, interquartile range; BMI, body mass index; MELD, Model for End-Stage Liver Disease; GAG-HCC, Guide with Age, Gender, HBV DNA, Core Promoter Mutations and Cirrhosis-HCC; CU-HCC, Chinese University-HCC; REACH-B, risk estimation for hepatocellular carcinoma in chronic hepatitis B; FIB-4, fibrosis-4; PAGE-B, Platelet Age Gender–HBV.

**Table 2 cancers-16-02521-t002:** Comparison of clinical and laboratory variables between patients who developed HCC and those who remained free of HCC.

Variable	Developed HCC N = 34		Remained Free of HCC N = 598			*p*-Value
	N	%	N	%	*x^2^*	Fisher’s Exact Test
Gender, male	31	91.2%	353	59.0%	13.944	**<0.001**
Active smoker	12	35.3%	174	29.1%	0.595	0.443
Alcohol abuse	15	44.1%	97	16.2%	17.171	**<0.001**
Family history of HCC	1	2.9%	14	2.3%	0.050	0.568
Phase of infection					34.460	**<0.001**
HbeAg (+) chronic infection	0	0.0%	4	0.7%		
HbeAg (+) chronic hepatitis	0	0.0%	38	6.3%		
HbeAg (-) chronic infection	4	11.8%	315	52.7%		
HbeAg (-) chronic hepatitis	30	88.2%	241	40.3%		
Cirrhosis	23	67.6%	52	8.7%	106.898	**<0.001**
Decompensated cirrhosis	6	17.6%	18	3.0%	18.866	**0.001**
Ascites	4	11.8%	20	3.3%	6.244	**0.035**
Esophageal varices	10	29.4%	27	4.5%	36.180	**<0.001**
Portal gastropathy	8	23.5%	18	3.0%	34.338	**<0.001**
Portal vein thrombosis	0	0.0%	1	0.2%	0.057	0.999
Received treatment	29	85.3%	176	29.4%	45.810	**<0.001**
IFN-based	12	35.3%	60	10.0%		
NUCs	17	50.0%	116	19.4%		
	**Median**	**IQR**	**Median**	**IQR**	**Z**	
Age	63.0	55.0–67.0	34.0	45.0–60.0	5.670	**<0.001**
BMI	26.1	23.9–26.9	26.1	23.5–28.7	0.258	0.797
Ht (%)	42.4	38.7–45.6	42.4	38.7–45.5	0.018	0.985
Hb (g/dL)	14.1	13.2–15.4	14.1	12.9–15.2	0.226	0.821
PLT (K/uL)	132.5	93.5–191.0	208.0	168.0–250.0	5.270	**<0.001**
INR	1.2	1.0–1.3	1.0	1.0–1.1	3.659	**<0.001**
SGOT (U/L)	51.0	39.0–93.0	29.0	21.0–50.5	3.974	**<0.001**
SGPT (U/L)	60.0	40.0–138.0	35.0	21.0–65.0	3.432	**0.001**
ALP (U/L)	125.0	98.0–210.0	89.0	64.0–139.5	3.555	**<0.001**
GGT (U/L)	84.0	38.0–131.6	21.0	14.0–38.0	5.518	**<0.001**
Total bilirubin (mg/dL)	1.0	0.7–1.5	0.7	0.5–0.9	3.044	**0.002**
Albumin (g/dL)	4.2	3.8–4.5	4.5	4.1–4.8	2.980	**0.003**
Creatinine (mg/dL)	1.0	0.8–1.2	0.9	0.8–1.0	2.097	**0.036**
K (mmol/L)	4.4	4.0–4.8	4.4	4.1–4.7	0.093	0.926
Na (mmol/L)	139.0	138.0–140.0	140.0	138.0–141.4	1.203	0.229
P (mg/dL)	3.1	2.6–3.4	3.3	3.0–3.8	0.803	0.422
Mg (mg/dL)	2.00	1.8–2.0	1.9	1.8–2.1	0.368	0.713
TSH (mIU/L)	1.3	0.6–2.7	1.2	0.8–1.8	0.201	0.841
HBV DNA (IU/mL)	14.0	0.0–39,000.0	27.0	0.0–17,000.0	0.132	0.895
CP score	5.0	5.0–6.0	5.0	5.0–7.0	0.917	0.359
MELD score	9.0	8.0–11.5	10.0	7.0–13.0	0.222	0.824
CU-HCC	19.3	16.5–21.5	3.0	0.0–20.0	3.878	**<0.001**
GAG-HCC	105.0	87.0–112.0	55.0	43.0–69.0	7.267	**<0.001**
REACH-B	9.5	8.5–12.0	7.0	4.0–9.0	3.907	**<0.001**
PAGE-B	20.0	17.5–22.0	10.0	6.0–16.0	7.339	**<0.001**
FIB-4	3.7	2.2–5.0	1.1	0.7–1.9	6.239	**<0.001**

Abbreviations: HCC, hepatocellular carcinoma; IFN, interferon; NUCs, nucleos(t)ide analogs; BMI, body mass index; HBV, hepatitis B virus; Ht, hematocrit; Hb, hemoglobin; PLT, platelets; PT, prothrombin time; INR, international normalized ratio; SGOT, aspartate transaminase; SGPT, alanine aminotransferase; ALP, alkaline phosphatase; GGT, gamma-glutamyl transferase; K, potassium; Na, sodium; P, phosphorus; Mg, magnesium; TSH, thyroid-stimulating hormone; CP, Child–Pugh; MELD, Model for End-Stage Liver Disease; GAG-HCC, Guide with Age, Gender, HBV DNA, Core Promoter Mutations and Cirrhosis-HCC; CU-HCC, Chinese University-HCC; REACH-B, risk estimation for hepatocellular carcinoma in chronic hepatitis B; PAGE-B, Platelet Age Gender–HBV; FIB-4, fibrosis-4.

**Table 3 cancers-16-02521-t003:** Univariate and multivariate Cox regression analysis for HCC development.

Variable	Univariate Analysis	HR (95% CI)	Multivariate Analysis	aHR (95% CI)
Age (years)	**<0.001**	1.080 (1.050–1.112)	**<0.001**	1.086 (1.037–1.137)
Gender, male	**0.003**	5.941 (1.812–19.472)	**0.003**	7.696 (1.971–30.046)
BMI	0.648	0.986 (0.930–1.046)		
Cirrhosis	**<0.001**	12.936 (6.289–26.610)	**<0.001**	21.239 (6.001–75.167)
HbeAg (+)	0.372	0.356 (0.037–3.434)		
Alcohol Abuse	**<0.001**	3.573 (1.762–7.245)	**0.016**	2.903 (1.222–6.897)
HBV DNA (IU/mL)	0.409	0.894 (0.685–1.166)		
Hb (g/dL)	0.869	0.991 (0.896–1.098)		
PLT (K/uL)	**<0.001**	0.988 (0.983–0.993)	NS	
INR	**<0.001**	5.316 (2.292–12.329)	NS	
SGOT (U/L)	0.347	1.001 (0.999–1.003)		
SGPT (U/L)	0.816	1.000 (0.999–1.002)		
Total Bilirubin (mg/dL)	**0.008**	1.358 (1.084–1.700)	NS	
Albumin (g/dL)	**0.008**	0.586 (0.395–0.870)	NS	
Creatinine	0.865	0.969 (0.675–1.392)		
Na	0.834	0.996 (0.963–1.031)		

Abbreviations: HR, hazard ratio; aHR, adjusted hazard ratio; BMI, Body Mass Index; Hb, Hemoglobin; PLT, Platelets; INR, International Normalized Ratio; SGOT, serum glutamic-oxaloacetic transaminase; SGPT, Serum Glutamate Pyruvate Transaminase; Na, sodium.

**Table 4 cancers-16-02521-t004:** Performance of HCC risk scores within total population and different subgroups.

Biomarker c-Statistic (95% CI), *p* Value	Total Population (N = 743)	Treated Patients (N = 213)	Untreated Patients (N = 530)	Patients with Cirrhosis (N = 104)	Patients without Cirrhosis (N = 639)
**CU-HCC**	0.714 (0.613–0.815), 0.002	0.672 (0.540–0.803), 0.022	0.751 (0.690–0.812), 0.386	0.407 (0.228–0.587), 0.361	0.559, (0.358–0.759), 0.624
**FIB-4**	0.748 (0.656–0.841), <0.001	0.635 (0.511–0.759), 0.071	0.907 (0.866–0.948), 0.161	0.244 (0.052–0.435), 0.012	0.654, (0.484–0.824), 0.197
**REACH-B**	0.762 (0.685–0.840), <0.001	0.583 (0.474–0.702), 0.267	0.984 (0.966–1.000), 0.009	0.565 (0.377–0.753), 0.523	0.727, (0.584–0.869), 0.057
**GAG-HCC**	0.895 (0.829–0.961), <0.001	0.846 (0.75–0.943), <0.001	0.959 (0.930–0.987), 0.114	0.599 (0.423–0.774), 0.330	0.816 (0.685–0.947), 0.008
**PAGE-B**	0.857 (0.791–0.924), <0.001	0.818 (0.728–0.909), <0.001	0.847 (0.780–0.914), 0.232	0.602 (0.412–0.792), 0.315	0.808, (0.708–0.908), 0.010

Abbreviations: GAG-HCC, Guide with Age, Gender, HBV DNA, Core Promoter Mutations and Cirrhosis-HCC; CU-HCC, Chinese University-HCC; REACH-B, risk estimation for hepatocellular carcinoma in chronic hepatitis B; PAGE-B, Platelet Age Gender–HBV; FIB-4, fibrosis-4.

**Table 5 cancers-16-02521-t005:** Performance metrics for each prognostic score in the total population of patients with HBV.

	Cut-Off	Sensitivity (95% CI)	Specificity (95% CI)	PPV (95% CI)	NPV (95% CI)	LR+ (95% CI)	LR- (95% CI)	Accuracy (95% CI)
**CU-HCC**	15.75	0.765 (0.588–0.893)	0.652 (0.613–0.690)	0.111 (0.092–0.134)	0.980 (0.964–0.989)	2.20 (1.77–2.73)	0.36 (0.20–0.66)	0.658 (0.620–0.695)
**GAG-HCC**	78.9	0.828 (0.642–0.942)	0.858 (0.826–0.886)	0.231 (0.186–0.281)	0.990 (0.976–0.995)	5.82 (4.48–6.57)	0.20 (0.09–0.45)	0.856 (0.826–0.884)
**REACH-B**	6.5	0.950 (0.751–0.999)	0.493 (0.440–0.546)	0.096 (0.084–0.109)	0.994 (0.963–0.999)	1.87 (1.62–2.16)	0.10 (0.01–0.69)	0.517 (0.465–0.569)
**PAGE-B**	15.5	0.906 (0.750–0.980)	0.749 (0.709–0.786)	0.185 (0.158–0.215)	0.992 (0.977–0.997)	3.61 (3.00–4.35)	0.13 (0.04–0.37)	0.758 (0.720–0.794)
**FIB-4**	2.11	0.781 (0.600–0.907)	0.781 (0.743–0.816)	0.184 (0.150–0.224)	0.983 (0.967–0.991)	3.57 (2.79–4.56)	0.28 (0.15–0.54)	0.781 (0.744–0.815)

Abbreviations: GAG-HCC, Guide with Age, Gender, HBV DNA, Core Promoter Mutations and Cirrhosis-HCC; CU-HCC, Chinese University-HCC; REACH-B, risk estimation for hepatocellular carcinoma in chronic hepatitis B; PAGE-B, Platelet Age Gender–HBV; FIB-4, fibrosis-4; PPV, positive predictive value; NPV, negative predictive value; LR, likelihood ratio.

## Data Availability

The data presented in this study are available on request from the corresponding author.

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
