# Peer review of "Predictive Risk Factors and Scoring Systems Associated with the Development of Hepatocellular Carcinoma in Chronic Hepatitis B"

_cancers, 2024, doi:10.3390/cancers16142521_

Round 1

Reviewer 1 Report

Comments and Suggestions for Authors

This is a well written articles with appropriate methodology.

Minor comments:

“Seven hundred forty-three” should be “Seven hundred thirty-three”

Abstract, line 22, authors should replace “In the multivariate analysis“ with “In the multivariable Cox regression analysis”.

In all Tables. Authors use different number of digits after comma, sometimes 3, sometimes 2 or 1.  They should use the same number of digits and not shorten. For example , when they use 2 digits, they should write 3.00 rather than 3.

Author Response

“Seven hundred forty-three” should be “Seven hundred thirty-three”

We would like to thank the reviewer for these comments. We have corrected accordingly as the reviewer suggested. 

Abstract, line 22, authors should replace “In the multivariate analysis“ with “In the multivariable Cox regression analysis”.

We have corrected accordingly as the reviewer suggested. 

In all Tables. Authors use different number of digits after comma, sometimes 3, sometimes 2 or 1.  They should use the same number of digits and not shorten. For example , when they use 2 digits, they should write 3.00 rather than 3.

We have corrected accordingly as the reviewer suggested. 

Reviewer 2 Report

Comments and Suggestions for Authors

The manuscript titled "Risk Factors and Prognostic Scores Associated with 2 Hepatocellular Carcinoma Development in Patients with 3 Hepatitis B Virus" analyzes the risk factors as predictors of HCC in hepatitis B viral infection patients. The authors access several parameters that affect hepatocellular carcinoma (HCC) development in Caucasian in Greece. However, several major concerns need to be addressed before publication:

Major concerns;

1.     Why was AFP, the standard marker for HCC, not included in the analysis?

2.     Why were patients with non-alcoholic fatty liver disease excluded, but not patients with alcohol abuse? Both non-alcoholic fatty liver disease and alcohol abuse are critical factors driving HCC development.

3.     The authors study baseline clinical parameters of patients that effect HCC development, but some patients received 149 anti-HBV treatment [interferon (IFN)-based or nucleos(t)ide analogs (NUCs)], which significantly impacts patient outcomes and results. In clinical research, baseline clinical parameters refer to the set of measurements and observations collected at the beginning of a study, before any intervention or treatment is administered. This might explain why HBV viral load was not associated with HCC development in your results.

4.     The number of included patients is inconsistent. For example, in the abstract line 18, seven hundred and forty-three CHB consecutive adults (n=632) are mentioned. The total number of patients in Table 1 is 632, but the total number of patients in Table 2 is 743. Could you clarify this discrepancy?

Additional points that need to be corrected:

- Line 20: Full name should be mentioned first.

- Line 32: Do not capitalize “hepatocellular carcinoma.”

- Edit “HBsAg” and “HBeAg” for consistency.

- Add full name of every abbreviation used in the tables.

- In Table 2: Do not use capital letters unless it is initial word, e.g., “Active smoker.” There appear to be two main contents in this table; should it be separated into two tables? Also, clarify if "Fisher’s exact test" in the last column refers to the p-value.

- AUC value should be added along with the tested factor in ROC curve.

- In Table 5, the value of PPV and NPV should be comparable. PPV indicates the likelihood that a positive test result correctly identifies the disease, which is crucial for deciding whether to proceed with further diagnostic tests or treatments. NPV indicates the likelihood that a negative test result correctly identifies the absence of the disease, which helps reassure patients and avoid unnecessary treatments. So both PPV and NPV should be high for accurate disease diagnosis.

- In Figure 4, GAG-HCC graph, the color of line graphs is too similar; they should be differentiated.

Author Response

  1. Why was AFP, the standard marker for HCC, not included in the analysis?

We greatly appreciate this comment. AFP was not included in the analysis considering that it is a diagnostic marker and not a predictive factor for HCC development.

  1. Why were patients with non-alcoholic fatty liver disease excluded, but not patients with alcohol abuse? Both non-alcoholic fatty liver disease and alcohol abuse are critical factors driving HCC development.

We would like to thank the reviewer for this comment. The percentage of patients with increased alcohol consumption was 17.7% in our study population. We specifically investigated the potential association between increased alcohol consumption and HCC development which was, subsequently, found significant in the multivariate analysis as an independent predictor. MASLD is difficult to diagnose in a retrospective study considering that we did not have histological data in the majority of patients neither fat quantification with MRI or CAP measurements (Fibroscan).

  1. The authors study baseline clinical parameters of patients that effect HCC development, but some patients received 149 anti-HBV treatment [interferon (IFN)-based or nucleos(t)ide analogs (NUCs)], which significantly impacts patient outcomes and results. In clinical research, baseline clinical parameters refer to the set of measurements and observations collected at the beginning of a study, before any intervention or treatment is administered. This might explain why HBV viral load was not associated with HCC development in your results.

We would like to thank the reviewer for this comment. You are right. We should not mention antiviral treatment as a baseline characteristic. The median time between first visit and treatment initiation was 3 months (range 0-213). Only 34 patients (16.6%) out of 205 treated patients started antiviral treatment at initial visit. None was receiving treatment before inclusion in the analysis. We have made appropriate changes in the manuscript as well as in the Tables.

  1. The number of included patients is inconsistent. For example, in the abstract line 18, seven hundred and forty-three CHB consecutive adults (n=632) are mentioned. The total number of patients in Table 1 is 632, but the total number of patients in Table 2 is 743. Could you clarify this discrepancy?

We would like to thank the reviewer for this comment. This was due to a type error from a previous manuscript editing

Additional points that need to be corrected:

- Line 20: Full name should be mentioned first.

We have corrected accordingly, as suggested

- Line 32: Do not capitalize “hepatocellular carcinoma.”

We have changed accordingly, as suggested

- Edit “HBsAg” and “HBeAg” for consistency.

We have edited the terms in order to maintain consistency

- Add full name of every abbreviation used in the tables.

We have added the full names of the abbreviations where missed

- In Table 2: Do not use capital letters unless it is initial word, e.g., “Active smoker.” There appear to be two main contents in this table; should it be separated into two tables? Also, clarify if "Fisher’s exact test" in the last column refers to the p-value.

We changed capital letters with small ones where appropriate as suggested. It is separated into 2 tables to distinguish between qualitative and quantitative variables considering that they are expressed with different statistical tests (that way seems to be more comprehensive for the readers). Does the reviewer want to change us the current format? Also, we confirm that Fisher’s exact test refers to the p-value.

- AUC value should be added along with the tested factor in ROC curve.

We have added the AUC values in the ROC curve as indicated.

- In Table 5, the value of PPV and NPV should be comparable. PPV indicates the likelihood that a positive test result correctly identifies the disease, which is crucial for deciding whether to proceed with further diagnostic tests or treatments. NPV indicates the likelihood that a negative test result correctly identifies the absence of the disease, which helps reassure patients and avoid unnecessary treatments. So both PPV and NPV should be high for accurate disease diagnosis.

Thank you for your insightful comments on Table 5. Indeed, an ideal test would yield high values for both PPV and NPV. However, the indices that were evaluated in our analysis demonstrated very high NPV but low PPV. This does not diminish the utility of the indices; rather, it underscores their practical application. These cost-effective and easily measurable parameters can effectively rule out patients who are not at risk. Importantly, while a positive test result does not confirm diagnosis, it does pinpoint individuals who require closer screening with more precise and costly diagnostic tools

- In Figure 4, GAG-HCC graph, the color of line graphs is too similar; they should be differentiated.

We changed the color of the lines as suggested

Reviewer 3 Report

Comments and Suggestions for Authors

The manuscript Pastras et al., entitled “Risk Factors and Prognostic Scores Associated with Hepatocellular Carcinoma Development in Patients with Hepatitis B Virus”. This manuscript provides information about the role of male sex, advanced age, increased alcohol consumption and baseline cirrhosis can be used as significant individual independent predictors for HCC development. The manuscript is well written, and the data are well presented and supports the context of the present study. This study provides interesting information for the scientific community.

Authors aimed to evaluate factors associated with liver cancer hepatitis B patients. In the past, several factors were already suggested to be associated with a cancer development in CHB patients. However, early studies either included only few factors or only predefined risk settings. The presented study included CHB patients with and without cancer. The introduction is clear and well structures and includes all important information needed. As authors included more variables than previous studies, they achieved some novel results, for example thrombocytopenia was not a significant independent prognostic factor for HCC development. Another interesting finding was the significant decline of the diagnostic performance of all HCC risk scores in the subgroup of cirrhotic patients. Authors mentioned important limitations and their conclusions are consistent with the evidence and arguments presented.

Comments on the Quality of English Language

is acceptable

Author Response

We would like to thank the reviewer for the kind comments

Reviewer 4 Report

Comments and Suggestions for Authors

In this study the authors assessed the risk factors asscoiated with hepatocellular carcinoma in patients with chronic HBV. The authors compared between HCC+, HCC- and then calculated the ROC associated with some selected risk factors such as PAGE-B, Reach-B, FIB- 238 4, CU-HCC, GAG-HCC

Major concerns:

1) Status of HBV vaccine is not mentioned and should be an important factor in whole study

2) Rationale of chosing these prognostic factors are not clear and values of these factors do not show high specificity and/or sensitivity.

3) What about cirrhosis in both groups?

4) Can these factor predict the mortality rate?

5) Can the authors use the prognistic factor to build up a scoring system

Comments on the Quality of English Language

moderate editing

Author Response

  • Status of HBV vaccine is not mentioned and should be an important factor in whole study

We would like to thank the reviewer for this comment. All patients included in the analysis had chronic hepatitis B (HbsAg positive); thus none of them were vaccinated against HBV

  • Rationale of chosing these prognostic factors are not clear and values of these factors do not show high specificity and/or sensitivity.

We assessed the prognostic significance of factors that have been already identified (or analysed) in previous studies/guidelines as potentially associated with HCC development. Furthermore, we analyzed the prognostic significance of these specific HCC scoring systems as these are validated and currently used worldwide to assess the risk for HCC development. The assessment of sensitivity and specificity of these scores was one of the aims of this study; some of these Sp/Se are rather high for some specific scores (i.e. PAGE-B).

  • What about cirrhosis in both groups?

In table 2 we have already provided information on the rate of cirrhosis in each group of patients (with/without HCC development).

  • Can these factor predict the mortality rate?

The aim of this study was to assess the predictive factors associated with HCC development and to compare different scoring systems. We did not assess survival data considering that this would lead off the readers from the purpose of this manuscript.

  • Can the authors use the prognistic factor to build up a scoring system

We would like to thank the reviewer for this comment. We could potentially create a new prognostic score although this was not the purpose of this manuscript. Considering the existence of many other predictive scoring models in the literature, some of them with high predictive performance, a new model would not add something novel in the existing knowledge. Furthermore, from a statistical perspective, the development of a new model necessitates a validation cohort, apart from a training one, in order to assess its performance independently. Thus, if a new model was created, another external cohort would be needed to validate its performance.

Round 2

Reviewer 2 Report

Comments and Suggestions for Authors

We thank for the authors for addressing all questions and concerns. This article can be accepted for publication.

Reviewer 4 Report

Comments and Suggestions for Authors

No further comments

Comments on the Quality of English Language

Language is fine